# The Effect of an Innovative Combination of Bergamot Polyphenolic Fraction and *Cynara cardunculus* L. Extract on Weight Gain Reduction and Fat Browning in Obese Mice

**DOI:** 10.3390/ijms25010191

**Published:** 2023-12-22

**Authors:** Saverio Nucera, Federica Scarano, Roberta Macrì, Rocco Mollace, Micaela Gliozzi, Cristina Carresi, Stefano Ruga, Maria Serra, Annamaria Tavernese, Rosamaria Caminiti, Annarita Coppoletta, Antonio Cardamone, Tiziana Montalcini, Arturo Pujia, Ernesto Palma, Carolina Muscoli, Francesco Barillà, Vincenzo Musolino, Vincenzo Mollace

**Affiliations:** 1Pharmacology Laboratory, Institute of Research for Food Safety and Health IRC-FSH, Department of Health Sciences, University Magna Graecia of Catanzaro, 88100 Catanzaro, Italy; saverio.nucera@hotmail.it (S.N.); federicascar87@gmail.com (F.S.); rocco.mollace@gmail.com (R.M.); micaela.gliozzi@gmail.com (M.G.); rugast1@gmail.com (S.R.); maria.serra@studenti.unicz.it (M.S.); an.tavernese@gmail.com (A.T.); rosamariacaminiti4@gmail.com (R.C.); annarita.coppoletta1@gmail.com (A.C.); tony.c@outlook.it (A.C.); muscoli@unicz.it (C.M.); 2Department of Systems Medicine, University of Rome Tor Vergata, 00133 Rome, Italy; francesco.barilla@uniroma2.it; 3Veterinary Pharmacology Laboratory, Institute of Research for Food Safety and Health IRC-FSH, Department of Health Sciences, University Magna Graecia of Catanzaro, 88100 Catanzaro, Italy; carresi@unicz.it (C.C.); palma@unicz.it (E.P.); 4Clinical Nutrition Unit, Department of Clinical and Experimental Medicine, University Magna of Græcia of Catanzaro, 88100 Catanzaro, Italy; tmontalcini@unicz.it; 5Department of Medical and Surgical Sciences, University “Magna Græcia” of Catanzaro, 88100 Catanzaro, Italy; pujia@unicz.it; 6Pharmaceutical Biology Laboratory, Institute of Research for Food Safety and Health IRC-FSH, Department of Health Sciences, University Magna Graecia of Catanzaro, 88100 Catanzaro, Italy; v.musolino@unicz.it; 7Renato Dulbecco Institute, Lamezia Terme, 88046 Catanzaro, Italy

**Keywords:** WAT, BAT, adipose tissue dysfunction, inflammation, oxidative stress, nutraceuticals, metabolic syndrome

## Abstract

Obesity is one of the world’s most serious public health issues, with a high risk of developing a wide range of diseases. As a result, focusing on adipose tissue dysfunction may help to prevent the metabolic disturbances commonly associated with obesity. Nutraceutical supplementation may be a crucial strategy for improving WAT inflammation and obesity and accelerating the browning process. The aim of this study was to perform a preclinical “proof of concept” study on *Bergacyn^®^*, an innovative formulation originating from a combination of bergamot polyphenolic fraction (BPF) and *Cynara cardunculus* (CyC), for the treatment of adipose tissue dysfunction. In particular, *Bergacyn^®^* supplementation in WD/SW-fed mice at doses of 50 mg/kg given orally for 12 weeks, was able to reduce body weight and total fat mass in the WD/SW mice, in association with an improvement in plasma biochemical parameters, including glycemia, total cholesterol, and LDL levels. In addition, a significant reduction in serum ALT levels was highlighted. The decreased WAT levels corresponded to an increased weight of BAT tissue, which was associated with a downregulation of PPARγ as compared to the vehicle group. *Bergacyn^®^* was able to restore PPARγ levels and prevent NF-kB overexpression in the WAT of mice fed a WD/SW diet, suggesting an improved oxidative metabolism and inflammatory status. These results were associated with a significant potentiation of the total antioxidant status in WD/SW mice. Finally, our data show, for the first time, that *Bergacyn*^®^ supplementation may be a valuable approach to counteract adipose tissue dysfunction and obesity-associated effects on cardiometabolic risk.

## 1. Introduction

Obesity occurs when an imbalance between energy intake and expenditure leads to changes in adipose tissue. Adipocyte hyperplasia and hypertrophy accompanying obesity are linked to impaired adipokine secretion and chronic inflammation, causing dysfunction in adipose tissue and disruption in systemic energy regulation [1,2]. Therefore, dysfunction in fat cells contributes significantly to obesity-related disorders like insulin resistance, diabetes, dyslipidemia, and an increased risk of heart disease [3,4], although the pathophysiological mechanisms involved in adipose tissue impairment remain to be better clarified.

Evidence exists that adipose tissue is no longer considered a simple deposit of energy substrates but works as an endocrine organ capable of producing a wide range of hormones and cytokines. These are involved in the metabolism of glucose (adiponectin, resistin), lipids (cholesteryl ester transfer protein (CETP)), inflammation (tumor necrosis factor α (TNFα), Interleukin-6 (IL-6), Interleukin-1 (IL-1)), fibrinolysis (through the modulation of the adipocyte-derived plasminogen activator inhibitor-1 (PAI-1)), the regulation of blood pressure (angiotensinogen, angiotensin II), and, finally, feeding behavior (leptin), thereby maintaining the body composition. Indeed, leptin is a peptide hormone exclusively expressed by adipocytes and essential for body weight regulation; rodents and humans that lack either leptin or the leptin receptor (LEPR) are obese, but also hyperglycemic and insulin-resistant [5,6,7]. Mammals generally have three types of adipose tissue: white adipose tissue (WAT), brown adipose tissue (BAT), and beige adipose tissue. WAT’s primary role is to store energy as triglycerides and release energy as fatty acids (FAs). In particular, evidence exists that WAT can be classified into two types: visceral white adipose tissue (vWAT) and subcutaneous white adipose tissue (scWAT) [8]. The first, which is highly metabolically active, releases FFAs into the portal circulation, contributing to various metabolic syndrome (MS) characteristics. The uncontrolled growth of WAT during obesity can cause fat to accumulate in non-adipose tissues, increasing the risk of developing insulin resistance and T2DM. The initial cause of insulin resistance and metabolic changes related to WAT expansion is the infiltration of adipose tissue macrophages (ATMs) and the subsequent increase in proinflammatory mediators [9]. Clinical studies also support the evidence that WAT from insulin-resistant and obese patients shows more macrophages with increased chemokine expression [10]. Moreover, it has been shown that FFAs released by hypertrophic adipocytes bind to TLR4, triggering inflammation through NF-κB activation [11]. Finally, it is known that the activation of TLR4 directly causes ATM to polarize into a constant activated state (M1), resulting in the secretion of TNFα, IL-6, and IL-1β at high levels. This was also confirmed by in vivo studies showing attenuated TNFα expression in TLR4 −/− mice fed a hyperlipidemic diet [12].

BAT seems to play a different role compared to WAT. Indeed, BAT consumes FAs and dissipates energy as heat, a process known as non-shivering thermogenesis. Increased energy expenditure may prevent excess weight gain and obesity development [13]. By increasing FA uptake and oxidation, BAT can lower triglyceride levels and mitigate the risk of atherosclerosis [14]. Based on studies in vitro and in vivo, the expression of uncoupling protein 1 (UCP1) in the inner membrane of mitochondria seems to represent the driver of the anti-atherogenic properties of BAT [15]. Indeed, the activation of BAT leads to the uncoupling of mitochondrial respiration and heat generation [16]. Originally, BAT was thought to be present in humans only until infancy, whilst PET scans recently identified significant deposits of brown adipocytes in adult humans. Research has indicated that activating BAT during adolescence leads to significant weight loss and reduced adiposity [17]. On the other hand, in situations of increased adaptive energy expenditure, beige or “brite” cells, similar to brown fat cells, emerge in WAT, particularly in scWAT depots. In particular, beige adipocytes have a thermogenic function that boosts energy expenditure, and adult WAT has the ability to undergo beige adipogenesis through various stimuli such as cold exposure, exercise, and agonists of the main adipogenic regulator PPAR-γ [18,19,20]. This process is called the browning of WAT and represents the critical mechanism for protection from fat-related cardiovascular disorders [21]. Thus, targeting adipose tissue dysfunction to promote WAT browning and BAT activation could significantly impact metabolism and combat obesity.

Nutraceuticals have been proposed as a potential solution for managing overweight and obesity [22]. Nevertheless, there is a lack of information regarding the impact of nutraceutical supplementation on the connection between WAT and BAT in obese subjects. Bergamot polyphenolic fraction (BPF) is a juice and albedo extract of bergamot recognized for its lipid-lowering properties due to its unique flavonoid and glycoside compositions [23,24,25,26,27]. In vivo data highlighted that BPF could prevent lipid profile alterations in rats fed a high-fat diet, thereby reducing oxidative stress and lipoprotein metabolism dysregulation [24]. The consequent restoration of acetyl-coenzyme A acetyltransferase (ACAT), lecithin cholesterol acyltransferase (LCAT), cholesteryl ester transfer protein (CETP), and paraxonoase-1 (PON1) activity is accompanied by apolipoprotein A1 (Apo A1) and apolipoprotein B (Apo B) level normalization [24]. In addition, recent evidence showed that bergamot polyphenolic extract treatment acts on metabolic balance and induces lipoprotein size re-arrangement, which is associated with reduced gut-derived lipopolysaccharide (LPS) levels, an effect associated with an improvement in gut microbiota as expressed by the modulation of the Gram-negative bacteria Proteobacteria, as well as Firmicutes and Bacteroidetes [28]. Indeed, gut microbiota modulation induced by high-fat diet feeding leads to short-chain fatty acid (SCFA) reduction and an increase in lipopolysaccharide production, which in turn facilitates fat storage and obesity [29]. Furthermore, it has been reported that BPF exerts anti-inflammatory activity in an in vivo NAFLD model due to its antioxidant activity and the inhibition of the JNK/p38 MAPK pathways [30]. Some polyphenols found in BPF have a specific feature that may protect against adipose tissue dysfunction linked to obesity, as shown by various experiments. Naringenin and hesperetin effectively inhibit TNFα-stimulated FFA secretion in 3T3-L1 adipocytes and mouse epididymal primary adipocytes by blocking NF-κB-mediated IL-6 transcription [31]. Moreover, naringenin treatment in epididymal adipose tissue reduces MCP-1 expression caused by a high-fat diet (HFD) by inhibiting JNK phosphorylation [32]. Additionally, naringenin reduced body weight and increased BAT UCP1 mRNA expression in HFD rats [33]. Recent studies reveal that naringenin boosts brown adipogenesis via PPARγ activation [34]. Similarly, the use of *Cynara cardunculus* L. extract (CyC), containing antioxidants like caffeic acid derivatives, flavonoids like luteolin, and sesquiterpenes like cynaropicrin, has demonstrated lipid-lowering and liver-protective effects in patients with high cholesterol levels [35]. Likewise, CyC counteracted hyperglycemia and hyperlipemia in HFD-fed rats. This effect was associated with a significant reduction in liver steatosis, thereby suggesting that the active ingredients found in the herbal extract could produce simultaneous beneficial effects in counteracting diet-induced metabolic imbalance and liver injury [36]. Finally, a recent study found that luteolin-enriched artichoke leaf extract can help reduce body weight and fat mass by activating lipogenesis in adipose tissue while increasing FA oxidation [37].

Given the pre-clinical and clinical evidence obtained by our research group on the beneficial individual effects of BPF and CyC in the prevention and treatment of disorders associated with metabolic syndrome, in this study, we assessed the potential benefits of *Bergacyn*^®^ (an innovative combination of BPF and CyC). In particular, since *Bergacyn*^®^ has already been proven to reduce oxidative stress and inflammatory biomarkers in patients with T2DM and NAFLD [38], we investigated its effect on adipose tissue dysfunction in hyperlipidemic-fed mice. 

## 2. Results

### 2.1. Effect of Bergacyn^®^ on Body Weight and Body Composition in a Diet-Induced Animal Model of Non-Alcoholic Fatty Liver Disease (DIAMOND)

Mice fed a Western diet with a high fructose–glucose water solution (WD SW diet) and treated with a vehicle showed a significant increase in body weight compared to normal chow diet and tap water (NC NW diet)-fed mice (Figure 1, red line; *p* < 0.001 vs. NC NW). The mice fed a WD SW diet and treated with 50 mg/kg/day of *Bergacyn^®^* showed a significant body weight gain reduction compared to the WD SW vehicle-treated group (Figure 1, green line). Likewise, the fat mass was significantly increased in the mice fed a WD SW diet and vehicle-treated compared to the mice fed an NC NW diet, and *Bergacyn^®^* administration significantly affected these parameters in the mice fed a WD SW diet (Figure 1).

### 2.2. Effect of Bergacyn^®^ on Adipose Tissue on DIAMOND Mice

The weight of eWAT was significantly increased in the WD SW-fed mice compared to the NC NW-fed mice; notably, the *Bergacyn^®^* administration significantly decreased eWAT in the mice fed a WD SW diet (Figure 2). 

The BAT weight was significantly increased in the mice fed a high-fat diet compared to the animals fed an NC NW diet (Figure 3A). In addition, the weight of BAT in the animals fed a WD diet and treated with *Bergacyn^®^* was higher than that of the WD SW vehicle group (Figure 3A). This evidence was supported by qualitative MRI imaging, which highlighted the same trend (Figure 3B).

### 2.3. Bergacyn^®^ Improved Total Oxidative Status

The effect of *Bergacyn^®^* on the total antioxidant status (TAS) was evaluated by a Randox assay. The results obtained showed that the TAS was significantly increased in the mice fed a WD SW diet and treated with *Bergacyn^®^* compared to the mice fed an NC NW diet and a WD SW diet, respectively (Figure 4).

### 2.4. Bergacyn^®^ Improved Dyslipidemia and Counteracted Liver Damage

The serum analyte assessment showed that the *Bergacyn^®^* administration counteracted the dyslipidemia and liver damage induced in the mice fed a WD SW diet. In particular, in the WD SW group, a significant increase in total cholesterol and LDL cholesterol (Figure 5A,B) was observed compared to the mice fed an NC NW diet; moreover, the mice receiving a WD SW diet showed an impairment in liver enzymes, with a significant increase in ALT levels (Figure 5D). The dyslipidemia and liver damage observed in the WD SW group were significantly reduced in the WD SW *Bergacyn^®^* mice (Figure 5).

### 2.5. Effect of Bergacyn^®^ on PPARγ, UCP-1, and NF-kB Expression in Adipose Tissues

The effect of *Bergacyn^®^* on PPARγ expression in BAT and WAT was evaluated. In brown adipose tissue, the expression of the protein was higher in the WD SW-diet-fed mice compared to the mice fed an NC NW diet, whereas the *Bergacyn^®^* treatment was able to reduce the protein level expression (Figure 6A). Moreover, the expression of UCP-1 was downregulated in the BAT of obese mice treated with the vehicle or *Bergacyn^®^* compared to the NC NW group (Figure 6B). On the contrary, in WAT lysates, the levels of PPARγ were lower in the mice fed a WD SW diet and treated with the vehicle compared to the NC NW group. The WD SW-fed mice treated with 50 mg/kg/day of *Bergacyn^®^* showed a restoration of PPARγ to the levels observed in the NC NW group (Figure 6C). Furthermore, the WAT lysate analysis showed that NF-kB levels were increased in the WD SW-diet-fed mice treated with the vehicle compared to the NC NW-fed mice. *Bergacyn^®^* (BRG) administration was associated with a significant reduction in NF-kB expression in the WAT of the DIAMOND mice (Figure 6D).

## 3. Discussion

Severe obesity is a major global public health issue linked to an increased risk of various diseases such as hyperlipidemia, cardiovascular diseases, hypertension, insulin resistance, and T2DM. Preventing metabolic disorders associated with obesity can be achieved by targeting adipose tissue dysfunction; on the other hand, nutraceutical supplementation may be a valid approach to improve WAT inflammation, combat obesity, and enhance the browning process, thereby reducing cardiometabolic risk [22,39,40]. Previous research from various groups, including ours, has indicated that a formulation called *Bergacyn*^®^, which combines bioactive molecules from *Citrus bergamia* and *Cynara cardunculus* L., has lipid-lowering, antioxidant, and anti-inflammatory properties that can improve multiple aspects of metabolic syndrome (MS) [38,41]. Our focus was on the effect of *Bergacyn*^®^ on adipose tissue dysfunction, the main initiator of MS, and on the proliferation and differentiation of WAT and BAT, which are regulated by a complex network of transcriptional factors, with PPARγ being a key player in adipogenesis, lipid metabolism, and insulin sensitivity [42]. In this work, we demonstrated that *Bergacyn*^®^ supplementation in obese mice fed with WD/SW reduced their body weight and total fat mass, improving plasma biochemical parameters, including total cholesterol and LDL. This confirms previous studies that show *Bergacyn*^®^ treatment improves lipid profiles. Furthermore, it has been demonstrated that diabetic mice fed an anthocyanin-extract-enriched diet showed an amelioration of the plasma glucagon-like peptide 1 (GLP-1) concentration, resulting in hyperglycemia reduction [43]. A similar effect was observed in an in vivo model of T2DM, showing that resveratrol was able to reduce glycemia through GLP-1 stimulation [44]. In addition, citrus polyphenols, such as rutin, eriocitrin, naringenin, hesperidin, and hesperetin, could inhibit Dipeptidyl Peptidase IV (DPP-IV) activity [45], thereby potentially increasing the half-life of circulating GLP-1 [46]. Thus, it is likely that *Bergacyn*^®^ may lead to body weight and hyperglycemia reduction through the same mechanisms. Our data showed that the reduction in WAT levels corresponded to an increase in BAT tissue weight, indicating the activation of a protective mechanism induced by *Bergacyn*^®^ that may be mediated by PPARγ. In addition, the trend of glycemia reduction observed in the WD SW group treated with *Bergacyn*^®^ compared to the vehicle group, although not significant, could possible be due to PPARγ downregulation. Recent discoveries indicate that diet-induced obesity causes insulin resistance in mice associated with BAT dysregulation [47]. Furthermore, as individuals age, there is a decline in both the mass and activity of BAT, leading to an effect on body fat accumulation, a loss of quality and quantity in skeletal muscle, glucose homeostasis dysregulation, and increased oxidative stress and inflammation [48,49,50,51].

On the other hand, *Bergacyn*^®^ was found to affect the “whitening” of BAT, a process known as the appearance of a WAT-like morphology, which indicates BAT degeneration [52]. In particular, thermogenin/UCP-1-mediated heat generation in the respiratory chain allows for the rapid oxidation of the substrate with little ATP production [53]. In our case, the decreased levels of UCP-1 in the vehicle group confirmed the occurrence of BAT dysfunction, which promotes tissue aging and activates the “whitening” process, as compared to the control group. Furthermore, *Bergacyn*^®^ was also found to be able to enhance BAT function by reducing PPARγ expression, an effect which contributes to counteracting BAT whitening. This finding aligns with evidence indicating that PPARγ acetylation worsens BAT whitening and contributes to metabolic dysfunction [54]. Instead, an inhibition of lipid accumulation in BAT was observed in PPARγ deacetylation-mimetic 2KR mice [55]. Conversely, the “browning” process can lead to a more oxidative phenotype in WAT, with the emergence of UCP1-containing mitochondria-rich beige cells [56]. This catabolic remodeling process involves multiple modulators of PPARγ binding activity. In particular, Sirtuin 1, also called the NAD-dependent deacetylase, triggers the brown adipogenic program by deacetylating PPARγ and recruiting PRDM16 [57]. 

Our results indicate that *Bergacyn*^®^ could restore PPARγ levels in mouse WAT on a WD/SW diet, potentially improving oxidative metabolism similar to the BAT phenotype, despite the absence of detectable UCP-1. Moreover, the upregulation of PPARγ demonstrates the anti-inflammatory and antioxidant properties of *Bergacyn*^®^ [38]. This is in accordance with data describing the ability of PPARγ agonists to reduce macrophage activation and decrease the expression of pro-inflammatory cytokines in obese mice [58,59]. To validate this, in the WAT of the WD/SW mice, *Bergacyn*^®^ inhibits the overexpression of NF-kB, the primary transcription factor for pro-inflammatory cytokines such as TNF-α and IL-6, which play a role in FFA secretion via adipocyte lipolysis and insulin resistance. In this context, dysregulated adipocytokine production from hypertrophic WAT is shown to play a role via increased NADPH oxidase activity and superoxide anion production while decreasing mRNA expression and the activities of antioxidant enzymes like CAT, SOD, and GPx [60]. In particular, the activation of PPARγ has been reported to positively impact adiponectin transcription and secretion, leading to increased fat oxidation and improved insulin sensitivity [61]. Dysfunction in adipose tissue leads to increased oxidative stress, resulting in the downregulation of adiponectin expression. Furthermore, patients with MS and obesity have shown low levels of TAS and high levels of serum C-reactive protein (CRP), indicating low-grade systemic inflammation [62]. Our findings indicate that *Bergacyn*^®^ supplementation decreases oxidative stress and improves adipose tissue function, enhancing the overall antioxidant systems in the WD/SW mice. This finding aligns with evidence suggesting that low TAS levels indicate higher oxidative stress and/or damage [63]. Examining the total antioxidant capacity is crucial, as the interaction between enzymatic and non-enzymatic antioxidant components helps protect against reactive oxygen and nitrogen species damage for each component [64]. Thus, the enhancement of the redox state through reduced oxidative stress and an increased antioxidant system may be a crucial mechanism behind *Bergacyn*^®^’s ability to promote a reduction in weight gain and fat mass. Furthermore, studies have shown that BPF, sesquiterpenes, and polyphenols of CyC, whether used alone or in combination (BPF+CyC), can enhance liver function by regulating oxidative stress and inflammation in both in vivo and clinical research [30,38,41]. Elevated ALT levels, a gold standard marker for liver damage, are directly linked to the accumulation of visceral fat and an increased body mass index (BMI), leading to fatty liver accumulation and inflammation [65]. Furthermore, the effect of *Bergacyn*^®^ on the adipose tissue phenotype, serum cholesterol, and ALT is not mediated only by reducing weight gain but is also due to a reduction in oxidative stress and systemic inflammation, which are modulators of the rate of progression of atherosclerosis, which could have some role in the pathogenesis of metabolic syndrome and NAFLD [66]. 

In our study, we found that supplementing with *Bergacyn*^®^ led to significant improvements in the lipid profile, body weight, and fat mass of the WD/SW mice. Additionally, the serum ALT levels were significantly reduced. The choice to perform all experiments only on male mice derives from the evidence highlighting that they are more susceptible to gaining more weight and becoming obese when exposed to HFD and more prone to developing HFD-induced inflammation, which may exacerbate their metabolic imbalance, compared to female mice [67]. Given the promising results, it would be interesting to take into account the sex differences in future metabolic studies involving the evaluation of the protective effects of Bergacyn^®^. 

To sum up, our data validate that WD/SW causes changes in the lipid profile, weight gain, and increased fat mass, along with elevated oxidative stress and liver damage markers in DIAMOND mice. In addition, our data indicate, for the first time, that *Bergacyn*^®^ supplementation counteracts adipose tissue dysfunction. In fact, *Bergacyn*^®^ increased BAT mass and prevented whitening, an effect associated with weight reduction and enhanced browning via modulation of PPARγ/NF-kB signaling and through the potentiation of the antioxidant system. This implies that the innovative formulation of *Bergacyn*^®^ may be a valuable approach to improving inflammation in adipose tissue and other negative effects of obesity on cardiometabolic risk.

## 4. Material and Methods

### 4.1. Animals

Male DIAMOND (diet-induced animal model of non-alcoholic fatty liver disease) mice were purchased from Sanyal Biotechnology (Virginia Beach, VA, USA) and kept under standard laboratory conditions in a specific-pathogen-free animal facility and maintained at 22 ± 2 °C with an alternating 12 h light–dark cycle [30]. Two mice were maintained in each cage, and mouse handling and care were provided in accordance with recommendations from the mouse metabolic phenotyping facility. The experimental procedures were carried out in accordance with protocols approved by the Animal Care Department of the University Magna Graecia of Catanzaro, in line with the European Commission guidelines (Directive 2010/63/EU) for animals used for scientific purposes. 

### 4.2. Bergacyn^®^ Preparation and Dilution

*Bergacyn^®^*, provided by Herbal and Antioxidant Derivatives (H&AD) (Bianco, Reggio Calabria, Italy), was obtained and characterized for polyphenol content [41] and used co-grinded and micronized with bergamot albedo fibers 50/50% (*Bergacyn FF*) (Appendix A). *Bergacyn^®^* was diluted with water (vehicle) to a final concentration of 0.015 mg/μL. Animals were gavaged daily, without anesthesia, with a dose of 50 mg/kg of *Bergacyn^®^* plus bergamot fibers. 

### 4.3. Study Design

Mice (8–12 weeks old; mean weight 22.6 g) were fed a normal chow diet (NC, Harlan TD.2019—protein: 19.2% by weight; carbohydrates: 44.9% by weight; fat: 9% by weight) and tap water (NW) or a high fat/high carbohydrate diet (Western diet, WD, Harlan TD.88137—protein: 17.3% by weight; carbohydrates: 48.5% by weight; fat: 21.2% by weight) and with a high fructose–glucose water solution (SW, 23.1 g/L d-fructose + 18.9 g/L d-glucose) for 28 weeks, ad libitum [30]. WD SW mice were treated with a vehicle (*n* = 10) or 50 mg/kg/day of *Bergacyn^®^* (*n* = 10) through gastric gavage, once daily, for 12 weeks, starting from week 16. NC NW mice (*n* = 10) did not undergo any treatment.

Body weight was measured the day before the diet regimen (baseline) started and once a week for the entire duration of the experimental period.

At sacrifice, mice were made to inhale isoflurane before being euthanized through cervical dislocation. Intrascapular brown (iBAT), epididymal white (eWAT), and inguinal white (iWAT) adipose tissue were collected, weighted, and immediately frozen in liquid nitrogen and stored at −80 °C for further analysis.

### 4.4. Body Composition Analysis

EchoMRI-700TM (Echo Medical System, Houston, TX, USA) was used to perform nuclear magnetic resonance spectroscopy and to assess total body fat, lean mass, and body fluids. The determination of active nuclei in the tissues, through magnetic resonance, was carried out to analyze the body structures. 

### 4.5. Magnetic Resonance Imaging

Anesthesia was performed using 5% isoflurane in oxygen (2 L/min) for a period of 2 min, and until the end, with 2% isoflurane in oxygen (2 L/min). During the scan, a multiparameter monitoring system was used for vital signs such as heart rate, respiration, and body temperature.

The images were acquired with a Bruker Pharmascan 70/16 US 7.0 Tesla MR through a total body transmitter–receiver coil. Tesla 7 imaging sequences are T2 Turbo RARE with relaxation time (TR) = 850 ms and echo time (TE) = 27.0 ms.

### 4.6. Blood Collection and Analysis of Blood-Based Endpoints

Blood collection was performed via cardiac puncture from euthanized mice. Approximately 700 μL of whole blood for each mouse was collected into untreated Eppendorf tubes, allowed to clot for 30 min at room temperature, and then centrifuged for 15 min at 1500× *g* and 4 °C. The serum was collected and stored at −80 °C for subsequent analysis.

The serum levels of analytes and enzymes, including glucose (GLU), total cholesterol (CHOL), low-density lipoprotein cholesterol (LDL-C), alanine aminotransferase (ALT), and aspartate aminotransferase (AST), were assessed using an automatic chemistry analyzer, XL-640 (Erba Mannheim), and the following Erba liquid stable reagents: glucose (GLU 440, XSYS0012), total cholesterol (CHOL 440, XSYS0009), low-density lipoprotein cholesterol (LDL C 80, XSYS0044), alanine aminotransferase (ALT/GPT 330, XSYS0017), and aspartate aminotransferase (AST/GOT 330, XSYS0016).

The serum total antioxidant status (TAS) was evaluated by a spectrophotometric assay using the kit Randox (NX 2332) and the instrument Randox RX Monza (Randox Laboratories Ltd., Crumlin, UK). Specifically, the antioxidant assay involves ferryl myoglobin radical formation from metmyoglobin and hydrogen peroxide, which oxidizes the ABTS (2,2′-azino-bis (3-ethylbenzthiazoline-6-sulfonic acid)) to obtain the radical cation, ABTS+, a green soluble chromogen, spectrophotometrically determined at 600 nm. When a compound with antioxidant activity is added, the latter suppresses radical cation production in a concentration-dependent manner, with a proportional decrease in color intensity.

### 4.7. Protein Extraction and SDS-PAGE Western Blot Analysis

WAT lysates were obtained as follows: about 100 mg of WAT was homogenized in 400 μL of ice-cold lysis buffer (50 mM of Tris pH 8; 150 mM of NaCl; 1 mM of EDTA; 1% Triton X 100; 1% sodium deoxycholate; 0.1% SDS; 10 μL/mL of freshly added protease and phosphatase inhibitor cocktails) and incubated on ice for 1 h. Samples were centrifuged at 20,000× *g* for 20 min at 4 °C, the top lipid layer was discarded, and the supernatant was transferred to a new capped tube. The samples obtained were centrifuged again at 10,000× *g* for 10 min at 4 °C, and the supernatant was collected.

BAT lysates were obtained as described: about 100 mg of BAT was homogenized in 600 μL of ice-cold lysis buffer (50 mM of Tris pH 7.6; 150 mM of NaCl; 0.1% Triton X 100; 0.5% of sodium deoxycholate; 0.1% SDS; 10 μL/mL of freshly added protease and phosphatase inhibitor cocktails) and incubated on ice for 1 h. Samples were centrifuged at 14,000× *g* for 15 min at 4 °C, and the supernatant between the fat cake and the pellet with a 26/27G syringe was removed.

A volume of 20 μL of the supernatant was used to assess the total protein concentration by a BCA protein assay (Pierce™ BCA Protein Assay Kit, #23225, Thermo Scientific, Waltham, MA, USA) using bovine serum albumin (Quick Start Bovine Serum Albumin Standard, Bio-Rad #500-0206, Hercules, CA, USA) as a standard. Proteins were heat-denatured for 5 min at 95 °C in a sample loading buffer (500 mM of Tris/HCl, pH 6.8; 30% glycerol; 10% sodium dodecyl sulfate; 5% β-mercaptoethanol; and 0.024% bromophenol blue), and 20 μg of protein lysate was resolved by sodium dodecyl sulfate polyacrylamide gel electrophoresis and transferred to nitrocellulose membranes (Amersham Protan 0.2 µm NC 10600001, Little Chalfont, UK). Subsequently, membranes were blocked with Tris/HCl (pH 7.6) containing 0.1% Tween 20 and 5% BSA for 1 h and incubated overnight at 4 °C with shaking with the following primary antibodies: anti-NF-kB (Cell Signaling #8242, Danvers, MA, USA); anti-peroxisome proliferator-activated receptor γ (PPARγ) (Abcam ab41928, Cambridge, UK); anti-UCP1 (Cell Signaling #14670, Danvers, MA, USA); and anti-GAPDH (Abcam ab181602, Cambridge, UK). Membranes were then washed in Tris-buffered saline (TBS, pH 7.6) with 0.1% Tween-20 and incubated with horseradish peroxidase-conjugated secondary antibodies (anti-rabbit antibody Pierce #31460 or anti-mouse antibody Pierce #31430, Invitrogen, Carlsbad, CA, USA) for 1 h at RT with shaking. Bound antibodies were highlighted using the chemiluminescent kit (ECL WB Detection, GE Healthcare RPN210601819, Little Chalfont, UK) and immunoblot scanning and analyzed with an imaging system (UVITEC Imaging Systems, Cambridge, UK). Band quantification was carried out through ImageJ Fiji (version 2.3.0/1.53f, WS Rasband, NIH, Bethesda, MD, USA). 

### 4.8. Statistical Analysis

Data were analyzed with GraphPad PRISM 9.3.1 (GraphPad Software, Inc., La Jolla, CA, USA). The Shapiro–Wilk test was used to test normality. Normally distributed data were analyzed by one-way ANOVA followed by Tukey’s test; data not normally distributed were analyzed through the Kruskal–Wallis test followed by Dunn’s tests. The results are shown as means ± SEMs. A *p*-value < 0.05 was considered statistically significant.

## Figures and Tables

**Figure 1 ijms-25-00191-f001:**
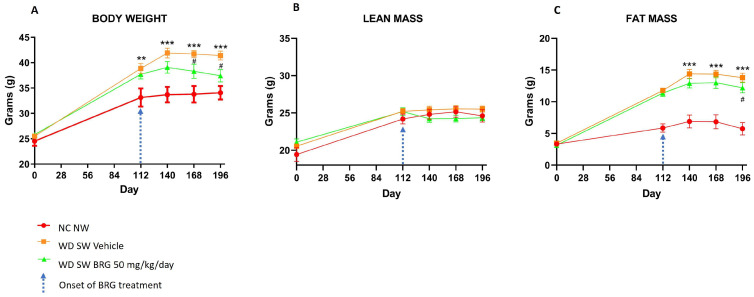
Effect of *Bergacyn^®^* on the body weight and body composition on WD SW-fed mice, quantified by nuclear magnetic resonance spectroscopy (EchoMRI-700TM). The body weight (**A**) and fat mass (**C**) of mice fed with a high-fat diet and treated with the vehicle increased significantly compared to that of the mice fed an NC NW diet. *Bergacyn^®^* (BRG) (50 mg/kg/day) treatment in WD SW-fed mice determined a significant reduction in body weight gain. No significant changes were observed in lean mass measurements (**B**). NC NW group (red line; *n* = 10); WD SW vehicle group (orange line; *n* = 10); WD SW *Bergacyn^®^* (BRG) group (green line; *n* = 10). The blue arrow on day 112 points to the time of BRG treatment onset. Results are expressed as mean ± SEM (**: *p* < 0.01; ***: *p* < 0.001 vs. NC NW; #: *p* < 0.05 vs. WD SW).

**Figure 2 ijms-25-00191-f002:**
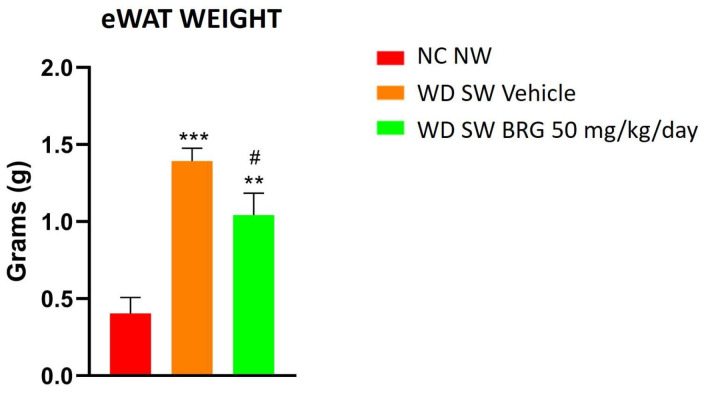
Effect of *Bergacyn^®^* on eWAT in WD SW-fed mice. The weight of eWAT increased significantly in mice fed with a high-fat diet and treated with the vehicle compared to mice fed an NC NW diet. *Bergacyn^®^* (BRG) (50 mg/kg/day) treatment in WD SW-fed mice determined a significant eWAT loss. NC NW group (red bar; *n* = 10); WD SW vehicle group (orange bar; *n* = 10); WD SW 50 mg/kg/day of *Bergacyn^®^* (BRG) group (green bar; *n* = 10). Data are expressed as the mean ± SEM. ***: *p* < 0.001 vs. NC NW; **: *p* < 0.01 vs. NC NW; #: *p* < 0.05 vs. WD SW.

**Figure 3 ijms-25-00191-f003:**
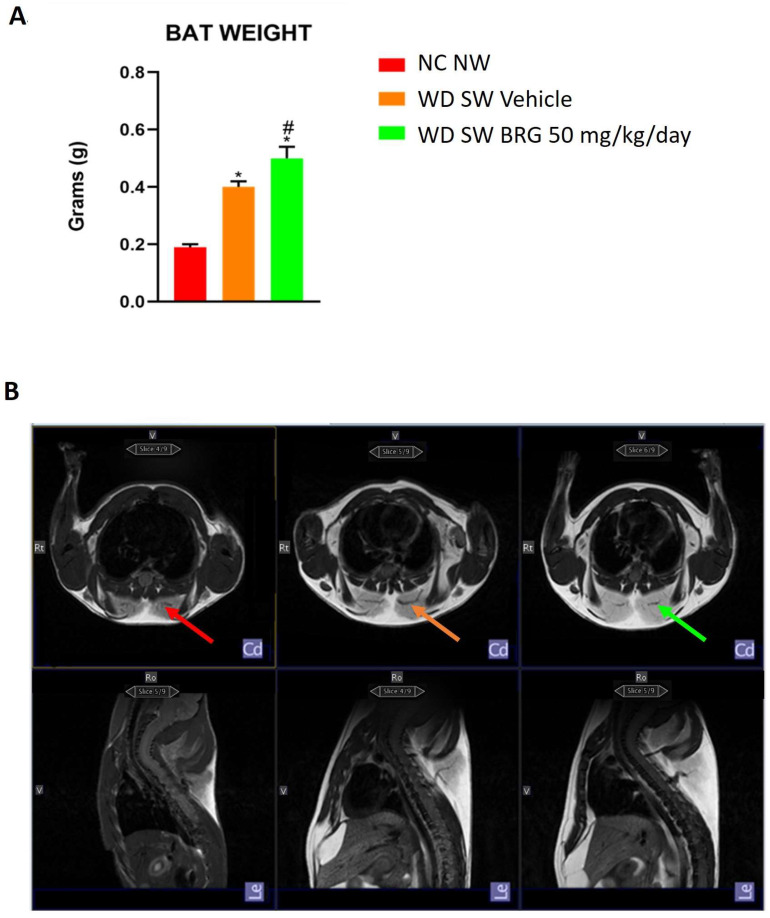
Effect of *Bergacyn^®^* on BAT tissue. (**A**) The BAT weight increased in mice fed a WD SW diet and treated with *Bergacyn^®^* (BRG) compared to mice fed an NC NW diet. Treatment with 50 mg/kg/day of *Bergacyn^®^* (BRG) increased even more the weight of the fat pad. The weight of WAT was similar in obese mice treated with vehicle or *Bergacyn^®^* (BRG), and this was higher compared to the NC NW animals. NC NW group (red; *n* = 10); WD SW vehicle group (orange; *n* = 10); WD SW 50 mg/kg/day of *Bergacyn^®^* (BRG) group (green; *n* = 10). (**B**) Representative in vivo axial (**up**) and sagittal (**down**) 7T magnetic resonance images (T2-weighted) of the mouse brown adipose tissue (BAT) of the three different experimental groups (see the arrows in red, orange, and green, respectively). NC NW group; WD SW vehicle group; and WD SW 50 mg/kg/day of *Bergacyn^®^* (BRG) group (from left to right). Data are expressed as the mean ± SEM. * *p* < 0.05 vs. NC NW; # *p* < 0.05 vs. WD SW vehicle.

**Figure 4 ijms-25-00191-f004:**
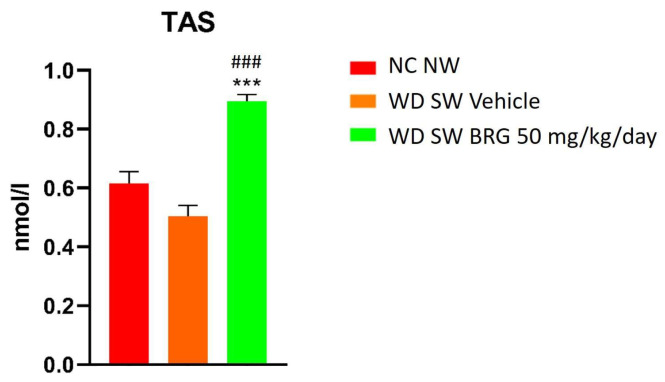
Effects of *Bergacyn^®^* on serum total antioxidant status (TAS). TAS was assessed at the end of the study in the serum of the NC NW (red bar; *n* = 10), WD SW vehicle (orange bar; *n* = 10), and WD SW *Bergacyn^®^* (BRG) 50 mg/kg/day (green bar; *n* = 10) mice. Data are expressed as the mean ± SEM. ***: *p* < 0.001 vs. NC NW; ###: *p* < 0.001 vs. WD SW.

**Figure 5 ijms-25-00191-f005:**
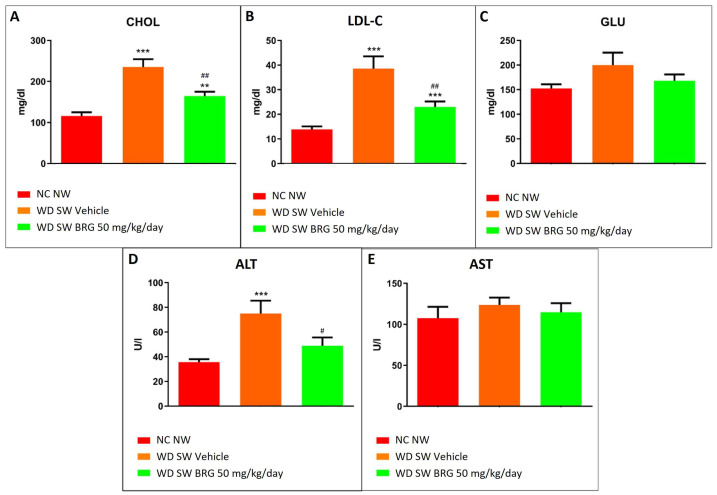
Effects of *Bergacyn^®^* on serum biochemical parameters. Different analytes, (**A**) CHOL, (**B**) LDL-C, (**C**) GLU, (**D**) ALT, (**E**) AST, were assessed at the end of the study in the serum of NC NW (red bar; *n* = 10), WD SW (orange bar; *n* = 10), and WD SW *Bergacyn^®^* (BRG) 50 mg/kg/day (green bar; *n* = 10) mice. (**A**): Serum cholesterol levels Data are expressed as the mean ± SEM. ***: *p* < 0.001 vs. NC NW; **: *p* < 0.01 vs. NC NW; ##: *p* < 0.01 vs. WD SW; #: *p* < 0.05 vs. WD SW. CHOL: cholesterol; LDL-C: low-density lipoprotein cholesterol; GLU: glucose; ALT: alanine aminotransferase; AST: aspartate aminotransferase.

**Figure 6 ijms-25-00191-f006:**
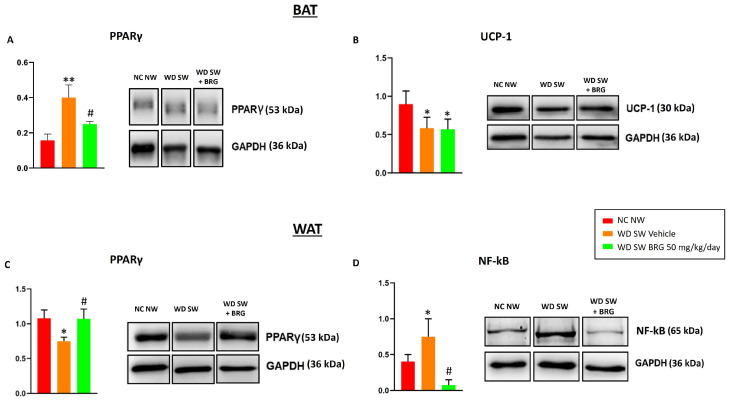
Expression of PPARγ, UCP-1, and NF-kB in adipose tissue. (**A**) The PPARγ expression in BAT was reduced in the WD SW group treated with 50 mg/kg/day of *Bergacyn^®^* (BRG) compared to the WD SW vehicle group. Furthermore, in the WD SW group, the expression of PPARγ increased significantly compared to the NC NW group. (**B**) The expression of UCP-1 in BAT was significantly reduced both in the WD SW group and in the WD SW group treated with *Bergacyn^®^* (BRG) compared to the NC NW group. (**C**) The expression of PPARγ in WAT was increased in the WD SW group treated with 50 mg/kg/day of *Bergacyn^®^* (BRG) compared to the WD SW vehicle group, restoring levels similar to those of the NC NW group. Furthermore, in the WD SW group, the expression of PPARγ decreased significantly compared to the NC NW group. (**D**) The expression in the WAT of NF-kB was significantly increased in the WD SW animals treated with the vehicle and reduced by the treatment with *Bergacyn^®^* (BRG). NC NW group (red bar; *n* = 10); WD SW vehicle group (orange bar; *n* = 10); WD SW BRG group (green bar; *n* = 10). Results are expressed as mean ± SEM * *p* < 0.05 vs. NC NW; **: *p* < 0.01 vs. NC NW; # *p* < 0.05 vs. WD SW vehicle. PPARγ: peroxisome proliferator-activated receptor gamma; UCP-1: uncoupling protein 1; NF-kB: Nuclear factor kappa-light-chain-enhancer of activated B cells.

## Data Availability

The data presented in this study are available upon request from the corresponding author.

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
