# Peer review of "The Effect of an Innovative Combination of Bergamot Polyphenolic Fraction and Cynara cardunculus L. Extract on Weight Gain Reduction and Fat Browning in Obese Mice"

_ijms, 2023, doi:10.3390/ijms25010191_

Round 1

Reviewer 1 Report

Comments and Suggestions for Authors

Nucera et al. present work where treatment with Bergacyn reduces obesity and a variety of inflammatory and oxidant stress markers. It is well written and straightforward. Minor comments included below largely focus on insuring that the work is optimized for reader ease and that definitions are defined prior to the methods or actually in the figures and legends. Also, the authors are asked to consider merging some of the single-panel figures into others. Overall, these concerns are minor and the author’s should be commended on their work.

1What exactly is Bergacyn? It’s not mentioned in the abstract or introduction what exactly it contains and this should be mentioned prior to the methods.

2 WD and SW are not defined prior to the methods.

3Consider putting the mouse numbers per group in the Figure legends and also include how fat was quantified.

4 Given that Figure 2 is just one panel, consider merging Figure 2 and 3 for ease of reading. Also, since it’s possible most readers won’t be familiar with MRI imaging, please highlight or put arrows showing relevant points showing the trend. Was there not enough N to quantify the MRI scans?

5In the text, legends and figures there’s some variance in the BRG being mg/kg/day or mg/Kg/day. Please change to mg/kg/day for consistency.

6Define TAS in the text, figure and legend. 

7Figure 5 needs panel letters (A, B, C, etc.) and also the titles defined in the legend. Consider adding letters to other Figures as well.

8Figure 6 needs the ladders included on the blots for expected MW.

9Figure 7 can be merged with Figure 6. There’s also a red squiggly line under NFkB, as well as a Nf-kB in the title. Change for consistency.

1Please note on Figure 1 where BRG was initiated.

1Just an FYI, the following articles PMID 36924940 and 35471692 are fairly current and relevant review articles to the topic.

Reviewer 2 Report

Comments and Suggestions for Authors

The aim of this research was to examine the effect of Bergacyn, the  combination of Bergamot Polyphe-2 nolic Fraction and Cynara cardunculus L. extract, on the development of obesity in mice fed high calorie diet. The results show that Bergacyn reduced body weight gain and fat mass gain in mice fed the Western diet, decreased the amount of epidydimal white and increased the amount of brown adipose tissue. Western diet itself had no effect on serum total antioxidant status but it was improved by Bergacyn administration. Western diet increased serum total and LDL-cholesterol levels as well as the activity of alanine aminotransferase and these markers were reduced by Bergacyn. The expression of PPAR-gamma and UCP-1 in BAT was increased and decreased, respectively, by Western diet. Bergacyn reduced the expression of PPAR-gamma but had no effect on the expression of UCP1. In WAT, the expression of PPARgamma was reduced by WEstern diet and normalized by Bergacyn. Bergacyn also reduced the expression of NF-kappaB in WAT of Western diet-fed mice. Together, the results show the beneficial effect of Bergacyn on metabolism and adipose tissue phenotype.

The results are of interest, however, there are also important concerns to be addressed.

1)     Lines 59-65: the section about the roles of specific adipokines oversimplifies their activity. For example, leptin not only regulates food intake but also glucose and lipid metabolism as well as inflammatory reaction. PAI-1 regulates fibrinolysis rather than coagulation.

2)     The major limitation of this study is that only one therapeutic approach was used. It would be reasonable to compare the effect of Bergacyn with that of its individual components.

3)     The composition of both diets (percent of calories provided by carbohydrates, fat and proteins) should be specified.

4)     Line 362: “low-density lipoproteins” should be corrected to “low-density lipoprotein cholesterol” since the amount of cholesterol contained in LDL rather than of LDL itself was measured.

5)     Line 133: it is stated that Bergacyn induced body weight loss which is the overinterpretation of the findings. According to Fig. 1 Bergacyn reduced weight gain rather that induced weight loss.

6)     What mechanism is responsible for the effect of Bergacyn on body weight gain? Did you measure food intake and/or energy expenditure?

7)     Is the effect of Bergacyn on adipose tissue phenotype, serum cholesterol and ALT mediated only by reducing weight gain? Could these effects be reproduced by other weight-reducing strategies such as pair-feeding or are there any weight gain-independent effects of Bergacyn?

8)     According to figure 5, neither Western diet nor Bergacyn had any significant effects on glucose levels. Thus, the discussion about its glucose-lowering effect is not supported by the results.

9)     All experiments were performed only in male mice which is the significant limitation of this study. This issue should be discussed as the effect could be different in female mice.

Round 2

Reviewer 2 Report

Comments and Suggestions for Authors

The manuscript has been revised according to the reviewers' comments. I have no further concerns.